# Micromanipulation System for Isolating a Single *Cryptosporidium* Oocyst

**DOI:** 10.3390/mi11010003

**Published:** 2019-12-18

**Authors:** Hamish Penny, David T. S. Hayman, Ebubekir Avci

**Affiliations:** 1Department of Mechanical and Electrical Engineering, Massey University, Palmerston North 4410, New Zealand; hamishrpenny@hotmail.com; 2School of Veterinary Science, Massey University, Palmerston North 4410, New Zealand; D.T.S.Hayman@massey.ac.nz

**Keywords:** micromanipulation, single cell analysis, image processing, prewitt operator, *Cryptosporidium*

## Abstract

In this paper, an integrated system for contact micromanipulation of *Cryptosporidium* oocysts is presented. The system integrates five actuators and a partially automated control system and contacts the oocyst using a drawn glass end effector with tip dimensions of 1 μm. The system is intended to allow single cell analysis (SCA) of *Cryptosporidium*—a very harmful parasite found in water supplies—by isolating the parasite oocyst of 5 μm diameter in a new environment. By allowing this form of analysis, the source of *Cryptosporidium* can be found and potential harm to humans can be reduced. The system must overcome the challenges of locating the oocysts and end effector in 3D space and contact adhesion forces between them, which are prominent over inertial forces on this scale. An automated alignment method is presented, using the Prewitt operator to give feedback on the level of focus and this system is tested, demonstrating alignment accuracy of <2 μm. Moreover, to overcome the challenge of adhesion forces, use of dry and liquid environments are investigated and a strategy is developed to capture the oocyst in the dry environment and release in the liquid environment. An experiment is conducted on the reliability of the system for isolating a *Cryptosporidium* oocyst from its culture, demonstrating a success rate of 98%.

## 1. Introduction

Many robotic systems of various types can operate and manipulate objects on the micrometer scale (commonly called micromanipulation). To deal with the unique challenges that become more prominent when objects become smaller, systems have had to become smarter, solving problems that on the macro scale would be relatively simple. In recent years, there has been significant research into developing the capabilities and applications of micromanipulation systems [1]. In particular, there is an interest in isolating single biological cells for single cell analysis (SCA) [2]. SCA is the gathering of information on the single cell level as opposed to a whole population, providing additional insight into the genetic, epigenetic, spacial, proteomic and lineage background of a cell type [3].

Common approaches for micromanipulation can be split into two main categories; non-contact and contact methods. Non-contact methods employ phenomena such as magnetism [4], light-dispersal from a laser [5], or fluid flow [6] for manipulation. Although some optical micromanipulation systems are capable of achieving manipulation of objects down to 1 μm diameter [7], there are difficulties in transporting an object from one environment to another due to a limited range of motion. These optical systems are also known to cause damage to biological cells when the laser is in contact with the cell [8]. The most common method for single cell isolation is the non-contact method of fluorescence activated cell sorters (FACS) [1]. FACS uses cell properties to detect and isolate cells of interest, and has proven very high throughput for some cell isolation tasks [9]. Some FACS systems claim the ability to isolate cells down to 5 μm diameter [10,11], but the cells must be isolated within the same liquid environment.

Contact methods use manufacturing techniques capable of creating physical manipulators with micro-scale features to physically contact and move objects [12]. Contact methods typically have a much larger range of motion (many times greater than the diameter of the object being manipulated) and have the potential to move an object between environments. For instance, the possibility of transferring silicate crystals (10–100 μm) from air to liquid environment has been demonstrated using a contact approach [13]. One of the significant challenges all micro-robotic systems—both contact and non-contact—aim to solve is accurately locating objects within 3D space. Approaches ranging from atomic force microscopy [14] to multi-camera microscopes [15] have been used to achieve improved accuracy of 3D localization. For contact methods, there is an additional challenge with contact adhesion forces such as capillary forces, van der Waals forces, and electrostatic forces, which become prominent over inertial forces as scale becomes smaller [16]. Existing contact micromanipulation systems have demonstrated promising results with artificial spheres down to 3 μm [14].

Information gathered from SCA has been proven useful with malaria-causing parasites [17] and circulating tumor cells [18], improving understanding and treatment of these diseases. The challenge for microrobotics is to expand the range of cells that can be isolated in a suitable environment for this analysis, as many cells are too small for existing systems to successfully isolate [19]. Systems for contact manipulation of cells have achieved successful pick and place of biological cells as small as the 9 μm cyanobacteria cell [20], but have not demonstrated isolating these cells in a new environment.

The specific intention of the proposed system is to isolate a *Cryptosporidium* oocyst of 4.2 μm–5.4 μm diameter [21] in a new environment (from a dry environment to a liquid environment) for the purpose of SCA. This oocyst—typically found in faecally contaminated water supplies—has a hard egg shell containing four parasites inside and is immotile until hatching. Diarrhoea causes greater mortality than malaria, measles and AIDS combined and *Cryptosporidium*, for which there is no vaccine, is the leading cause of diarrhoea morbidity and mortality in children under five years old globally [22]. *Cryptosporidium* oocysts are environmentally resilient and able to survive for months under suitable conditions [23] and are resistant to chlorination [24]. Most nation’s drinking water standards therefore include protozoal compliance criteria [24]. The standard *Cryptosporidium* water tests filter water to isolate oocysts and then stain and mount the oocysts on slides for epifluorescence microscopy. By developing a system for SCA, information can be gathered about the animal of origin of each parasite within each oocyst. This improves the ability to find the source of the oocyst in the water supply and allows the problem to be dealt with earlier—reducing harm to humans. The ultimate goal of the project is to investigate the difference between single oocysts at the genome level through SCA. The first part of this task is isolating a single oocyst in a new environment. There should be only one oocyst in each new environment (micro-wells in this case). After the isolation phase is completed, the second part, obtaining DNA from these single oocysts using a different mechanism, takes place. The final step of this task is analysing and comparing the genomes of each oocyst [11]. The focus of the current study is to complete the first part of this task, which is isolating single oocysts in a new liquid environment where further analysis can be carried out.

To achieve manipulation suitable for this size object, image processing methodologies have been employed along with an actuation system with five degrees of freedom. Drawn glass end effectors with tip dimensions of 1 μm are used to physically contact and manipulate the oocyst. The end effector is aligned with the focus plane and a substrate using focus feedback with the Prewitt operator. An oocyst is then picked from a dry environment, replicating the surface conditions of the slides used for epifluorescence microscopy to identify contaminated water, using the significant adhesion forces at this scale to adhere it to the end effector. Automated motion moves it to a liquid micro-well where it is released bythe movement of water caused by surface tension and the capillary force when entering the liquid environment. To the best of our knowledge, this is the first contact micromanipulation system that demonstrates isolation of a parasite from its culture to a new environment.

The proposed contact micromanipulation system is particularly developed for the parasite oocyst isolation task. Moreover, it has potential to be used for the isolation of any similar microorganisms for DNA sequencing purposes. The three highlights of the proposed system are as follows:Manipulation of biological micro-objects particularly smaller than 10 μm by using contact micromanipulation methods are not common as it is hard to detect 3D position of the target object and align the end effector with the target object successfully for capturing purposes. In this work, the possibility of handling biological micro-objects as small as 5 μm is demonstrated.Manipulating a micro-object in the same liquid environment is a very common way to demonstrate the capability of micromanipulation systems. However, picking up a micro-object (particularly as small as 5 μm) from one environment and transferring it to another environment is an arduous task and scarce in the literature. We demonstrate this is possible and actually crucial for some complicated tasks, e.g., oocyst isolation for DNA sequencing.Many studies in the literature have attempted to mitigate adhesion forces during the release of the target object by developing complex mechanisms. In this work, we show the possibility of utilizing forces at micro-scale to capture the target object and to release it without using any complicated mechanism.

## 2. Materials and Methods

### 2.1. Overview

The approach is to use one glass end effector with a tapered tip of radius 0.5 μm and pick and release an oocyst using this end effector and a stepper motor driven actuation system described in Section 2.2. The control system automating alignment of the focus plane, end effector, and oocyst, and automating oocyst release is detailed in Section 2.3. Details of the capture-and-release method are provided in Section 2.3.4.

### 2.2. Physical System

To allow reliable operation, a physical system was developed with the intention of minimising vibration of the end effector. The system shown in Figure 1 has been designed with this intention, as well as allowing flexibility in actuation modes. It is made up of stepper motor actuated stages providing five degrees of freedom for actuation—separated into the following functions:x, y, z actuators for independent movement of the end effector. x and y actuation has an 11.5 mm range of motion, z actuator has a 5.45 mm range of motion (Figure 1).x and y actuators for movement of the substrate. These actuators have a 52.45 mm range of motion (Figure 1).

The following equipment is used in the physical system. The stepper motor actuators (Sigma-Koki, Tokyo, Japan: two TAMM40 for end effector x and y actuation, one OSMS40 for end effector z actuation, two HPS80 for substrate x and y actuation) are used in micro-stepping mode using two controllers (Sigma-Koki, Tokyo, Japan: SHOT304GS), with a resolution of 0.025 μm movement at the end effector and substrate per step. The integrated system gives the ability to control the end effector in the x, y, and z axes—utilised for the capture strategy—and the substrate in the x and y axes—utilised to allow relocation of the end effector and oocyst to the liquid environment without moving out of the field of view of the microscope. An automated focusing system is implemented to allow the end effector to stay in focus during automated release. This stepper-motor actuated system (Stepper Motor, Nanjing, China: 42BYGP40 and custom made driver) has a resolution of 0.012 μm after connection to the microscope fine-focus rotary knob. As shown in Figure 1, the actuator system is mounted on the microscope (Olympus IX71 inverted microscope, Olympus, Tokyo, Japan) plate, which is mounted on the vibration isolation table (Thorlabs, Newton, NJ, USA: T1220CH). Use of this table and careful design of the actuator system has been observed to reduce vibration to the point (that is not visible under the microscope, therefore, practically less than a micron) where it does not have any significant impact on performance. For the initial system testing and development, the artificial spheres (Cospheric, Santa Barbara, CA, USA: SiO${}_{2}$MS-1.67 and PMPMS-1.2) are used, simulating the oocyst and allowing unlimited initial experiments on objects in the target size range. A micropipette puller (Narishige, Tokyo, Japan: PC-100) is used to create the drawn glass end effectors. A camera (IDS uEye camera) for vision feedback is integrated into microscope system. The dimensions of the microscope image were calibrated using an Olympus objective micrometer.

### 2.3. Control System

An application is developed using LabVIEW software for control of the system.The IDS uEye camera is connected to the main PC via USB, with IDS drivers installed and the NI-IMAQ function palette used in LabVIEW to interpret and process the image information. Image feedback is used to derive focusing information for automated alignment and to provide feedback to the operator for the manual capture operation. Visual feedback for motor control is not implemented in the currently proposed system. The microscope lenses allow up to 60× zoom for the captured image. The two SHOT 304GS stepper motor drivers for actuation and custom driver for the focusing stepper motor are controlled via serial link to the main PC. The control system is used for manual and automated control of these actuators and for automated object identification and 3D location.

#### 2.3.1. Object and End Effector Identification

To find the location of the oocysts and end effector within the field of view, a series of filtering techniques are used, as shown in Figure 2 below. In the first image of Figure 2 salt crystallisation after drying is shown. This is explained further in Section 3.3. The resolution of the image is 2592 × 1944 pixels and the field of view is 179 × 134 micrometers, meaning 14.5 pixels per micrometer.

The original image is firstly filtered with a Sobel edge detection algorithm. To remove noise a grayscale threshold is applied, and the image is filtered using proper close, fill holes, and particle filtering based on area. This is achieved using LabVIEW IMAQ VI’s. To isolate oocysts in the image, an additional segmentation takes place. As shown in Figure 2, this finds the Euclidean distance from each point in the binary particle to the nearest edge using the Danielsson algorithm [25]. Where there is a local minimum, the particle is split into two or more separate particles using the watershed algorithm [26]. Therefore, if two or more objects are touching and cannot be separated using binary filtering, they can still be identified using the watershed algorithm. The end effector is found using the LabVIEW IMAQ Shape Match Tool VI that compares the binary filtered shape to a known template based on the position of the edges (with scale invariance). The location of the tip of the end effector is fed into the focusing function to allow alignment of the end effector and focus plane (as described in Section 2.3.2). A manually selected region of interest (ROI) in the image is searched for objects in the expected diameter range, and the size and location of the oocyst are shown.In the lower right image in Figure 2, the units for position (X and Y) and radius are micrometers and the score is based on circle detection in the LabVIEW IMAQ detect shapes VI. This extracts the curves from objects and compares these to a circle, computing a score between 0–1000.

#### 2.3.2. 3D Location Method

The 3D location method is able to automatically align the end effector tip with the substrate, as well as bring the oocyst into focus. The oocyst sits directly on the substrate when in the dry environment (air). The end effector is first aligned with the focus plane as shown in Figure 3. To prepare for this automated alignment, manual focusing is used to bring the focus to some point above the level of the end effector. The focus plane (thick dotted line) is then held constant and the tip of the end effector is incrementally raised in 0.5 μm intervals through this focus plane. The end effector is then returned to the point of peak focus (the highest point on the graph).The zero (start point) on the distance axis of Figure 3 below represents an arbitrary start position for the end effector below the focus plane. This position is set up by the human operator. The exact location of this point does not matter as during this alignment process it is the relative location of the point of peak focus (the focus plane) and the end effector that is used.

In Section 3.2, an experiment was carried out to test the accuracy of the alignment between the end effector and focus plane. For that experiment, accuracy is defined as the vertical distance between the focus plane and the tip of the end effector after alignment. To calculate this, the following approach is carried out. The interference between two end effectors was observed, with the second “passive” end effector placed at a known angle relative to the end effector being aligned. To begin, an additional vertical actuator was used to manually align the passive end effector with the focus plane. The “active” end effector being tested was aligned with the focus plane using the automated alignment method, then actuated laterally in increments of 1 μm until interference between the two end effectors was observed. Given a number of known values—including an angle of 18 degrees between the end effectors, 10 degrees between the end effectors and the horizontal plane, and a taper of one degree (tapered end of the end effector)—a mathematical relationship between the point of interference and the alignment error was computed and used for the accuracy calculation during the alignment experiment in Section 3.2. Details of accuracy calculation can be found in Appendix A.

After alignment with the focus plane, the end effector is aligned with the substrate and oocyst as shown in Figure 4. As the end effector is lowered, the normalised focus value for the end effector is constantly read; when it drops below a threshold value derived by experimentation, it is lifted back up by a predefined distanceof 5 μm, also derived by experimentation.The threshold focus value is required to be lower than the minimum focus value possible due to noise as the end effector and focus plane are lowered toward the substrate (in focus) and greater than the maximum possible focus value when the end effector is completely out of focus. There is a significant difference between this minimum and maximum (as demonstrated in Section 3.2), so the threshold is selected at an arbitrary value in this range based on observation. The 5 μm distance to raise the end effector is empirically derived, where the end effector and focus plane are lifted by a range of different distances that are adjusted until the end effector is in focus and has firm contact with the substrate. As the exact level of force of the end effector on the substrate is not observed to have an impact on results throughout the pick and place experiment, it has been decided that a theoretical model to define these parameters is not required.

In Figure 4, the focus plane is held level with the end effector as it slowly moves down to the substrate. As the end effector tip reaches the base of the substrate it is stopped by contact with the microscope slide, so moves out of focus (as focus plane keeps moving down), and the normalised focus value decreases. The end effector and focus plane are then lifted to be aligned with the substrate and oocyst.

#### 2.3.3. Control System Algorithm

The full algorithm used in the LabVIEW control system is shown in Figure 5. The flow chart represents the whole process for each oocyst capture, and repeats from the end back to the start. An advantage of returning to the original location of oocyst capture is often several oocysts are located within a short distance of each other. Each oocyst in a certain area of the culture can be isolated sequentially.

In Figure 5, the first two sections describe the automated alignment of the focus plane, end effector, and oocyst. In the final operation of the alignment with the oocyst, the focus plane is raised by 2 μm more than the end effector (which is raised by the “set value” of 5 μm). This causes the end effector to be touching the substrate, while the focus plane is level with the center of the oocyst. The end effector touching the substrate causes an improved ability to capture the oocyst. The details of capture and release of the oocyst are explained in the next section.

#### 2.3.4. Capture and Release Method

Figure 6 demonstrates capture and release processes. After the oocyst and end effector are aligned automatically, the oocyst is manually captured in the dry environment. Once the oocyst is adhered to the end effector, it is raised by 1000 μm and the substrate is actuated to bring the liquid environment (automatically relocating the desired micro-well) under the end effector with oocyst. The end effector is then lowered 800 μm into the liquid environment to release the oocyst on entry into the liquid using force induced by surface tension and capillary action. The end effector is then lifted back out of the liquid environment and to the original location of oocyst capture. It has been observed that the end effector must be lifted high enough to have the water release from the tip, rather than adhere due to the capillary force. Using a pipette the volume of water released on the slide was consistent, with a height of approximately 500 μm. To overcome the capillary force and release the end effector from the liquid environment the end effector was lifted to 1000 μm above the substrate. Finally, the substrate is actuated back to the original position of oocyst capture and the end effector is lowered 1000 μm back to the level of the substrate.

During the capture and release operations, adhesion forces including the van der Waals force, electrostatic force, and capillary force are used as part of the strategy. It has been demonstrated in the literature that when object dimensions are less than 100 μm, adhesion forces become more significant than gravity and inertial forces, and that the van der Waals force and electrostatic force increase as the surface area of an object in contact or near to the other object increases [27]. During oocyst capture it was observed that oocyst adhesion to the substrate was very high initially and reduced after the oocyst was dislodged by the end effector. There are two likely mechanisms causing this. Firstly, it was observed that salt crystallisation and possible liquid residue would remain around the site where the oocyst dried. Secondly, it is possible that—despite the oocyst being a very hard biological cell—the cell would deform slightly during the drying process and flatten against the glass substrate. Both of these would increase the surface area in contact with the oocyst and therefore the contact adhesion forces. Either one or both of these may contribute to the initial adhesion force for a given oocyst.

For oocyst capture, the strategy is to push the end effector under the oocyst to overcome these forces as shown in Figure 7. Once the initial forces adhering the oocyst to the substrate are overcome and the oocyst is dislodged, it is relatively easy to move the side of the end effector against the oocyst (to maximise surface area) and have it adhere.

To release the oocyst, surface tension and the capillary force are used. The end effector approaches the surface of the liquid environment from above (so the surface of the water is near horizontal) and is lowered vertically until contact with the liquid. Just before the point of contact, surface tension causes the water surface to deform slightly, building up elastic energy in the water surface. At the point of the end effector breaking through the water surface the capillary force causes the water to rise up the end effector rapidly. The inertia of the water has sufficient force on the oocyst to cause detachment from the end effector as shown in Figure 7. As the oocyst has a greater density than water it gradually falls to the bottom of the liquid environment due to gravity.

### 2.4. Statistical Analyses

Statistical analyses of trials were performed in R version 3.5.2 using the binom.test to estimate the 95% confidence intervals of the probability of success in a Bernoulli experiment and Pearson’s Chi-squared test with Yates’ continuity correction to the differences between counts.

## 3. Results

### 3.1. Preparing Cryptosporidium Sample

A 0.17 mm thick microscope slide is first prepared by washing using ethanol. A sample of *Cryptosporidium* oocysts in water is then placed on the slide using a pipette and left in conventional lab environment up to three hours to evaporate the surrounding water so oocysts can be picked up in air.

### 3.2. Alignment Experiments

An experiment on the effectiveness of the end effector alignment strategy was conducted to compare the relative effectiveness of different focusing functions. The different functions considered include; Prewitt, gradient, differentiation, convolution, Sobel, sigma, and Roberts. These functions were chosen as they are commonly used for similar applications and were readily available for use in LabVIEW.

The reliability of each operator was tested by repeating up and down strokes through the focus plane. A 30 trial experiment was conducted to test the level of error of the alignment based on contacting the end effector tip with the tip of a stationary second end effector and observing interference.

Using the range of edge detection operators that have been mentioned above, the end effector was panned up through the focus plane and down again to return to its original position. The results of the first half (the pan up) for each operator are shown in Figure 8. The edge detection operators give a greyscale image output in the selected image region with pixel intensity values ranging from 0 to 255 representing the intensity of the edge at that pixel. The “level of focus” is the average grey level in the selected image region, representing the average edge intensity.

The two important criteria for an effective operator are based on the requirements of the control system for this particular task, but are similar to the requirements suggested in other similar research [28,29]:A clearly defined peak—it is important that the operator has one clear main peak, and that this peak is steep and narrow enough to find the center.An accurate peak location—to use the alignment method for micro-manipulation an accuracy of at least 2 μm is required. Therefore, the peak must be in the same place each time.Given the typical oocyst diameter is 5 μm and perfect alignment with the oocyst is not required (as our aim is to scrape an oocyst from the substrate), a 2 μm accuracy is sufficient for the pick and place operation.

From Figure 8, it can be observed that the Prewitt operator has the most clearly defined peak, and that all of the operators have a peak in a very similar position (at a distance of 20 μm from the zero position). Based on the results, the Prewitt operator was chosen as the optimal edge detection method. Subsequently an experiment of 30 trials was conducted to observe the accuracy of end effector tip alignment, using the experiment method described in Section 2.3.2. Based on the calculated results, the trial was categorised as having <2 μm accuracy or <1 μm accuracy (those in the <1 μm category would appear in both).

As shown in Table 1, the results indicate that the alignment method is consistent in achieving a 2 μm accuracy and this is significantly better than at the 1 μm accuracy ($\chi^{2}$ = 9.6, *p*-value = 0.002).

### 3.3. Pick and Place Reliability Experiments

To test the overall reliability of the system for the oocyst isolation task, a 50 trial experiment was conducted. Each trial consisted of:The automated system aligning the end effector and focus plane with the oocyst.The oocyst being manually picked from the dry substrate using the end effector.The end effector and oocyst automatically moving to the desired location, releasing the oocyst into an aqueous environment, and returning to the point it was initially picked from.

If all of these steps were successful and the oocyst is released on the first pass into the liquid environment, the trial was considered a success. In addition, the time required for each task was observed based on recorded video footage of all 50 trials.

A 50 trial experiment for pick and place task was conducted, yielding the results shown in Table 2.

The single failure was observed when the oocyst remained on the end effector after entering the liquid environment. An observation of this trial suggested the oocyst shape caused higher than normal contact area with the end effector, so contact adhesion forces were heightened. It could be expected that this form of failure would happen occasionally, but the high overall success rate of 98% suggests that this will be rare.

In Figure 9, the end effector is aligned with the focus plane then with the oocyst. Successful capture takes place by the process described in Section 2.3.4. There are salt crystals remaining around the oocyst site, as the liquid environment was phosphate-buffered saline solution with 8 g/L NaCl (along with lower concentrations of KCl, Na_2_HPO_4_ and KH_2_PO_4_) and when it dries salt crystals remains, which is a common observation in the dried environment. The end effector is then raised and the substrate is actuated to bring the micro-well close to the end effector. The oocyst is released into this liquid environment.

The time taken for each task was observed for all 50 samples in recorded video footage and details can be found in Table 3. In this table, σ represents the standard deviation.

The average total time required of 218 s—or 3 min and 38 s—is low enough to allow several oocysts to be isolated in separate environments within a reasonable time frame (an hour or less), so is suitable for the application. As can be seen in Table 3, the manual capture time can vary more than the other semi-automated tasks. The average duration for manual capture was 41 s and the maximum duration was 270 s, with four trials that took more than 90 s. For each of these the increased time was due to the difficulty in overcoming a high contact adhesion force between the oocyst and the substrate. This is likely to be caused by high contact area due to abnormal formation of salt crystals or oocyst deformation in the drying process (as described in Section 2.3.4). A single experienced operator completed all trials. A number of tests (50 trials) were conducted by the operator prior to the final experiment, so the operator was already reasonably proficient. The first 17 trials were conducted on the first day and the remaining 33 on the second day to reduce the chance of operator tiredness.

In Figure 10, the capability of the system for isolation of oocysts in separate environments is demonstrated. Four single oocysts are captured in the dry environment, then transported and released into the selected micro-wells (liquid environment). In these four trials the end effector was only lifted 100 μm above the level of the substrate and entered the water where the water surface had a greater angle. The successful release in these trials suggests the release mechanism does not rely on the angle of entry into the liquid environment. Details can be seen in the the Appendix A.

## 4. Discussion

### 4.1. Alignment Method

The comparison of focusing functions indicates that the optimal function for aligning the focus plane and end effector using auto-focusing is the Prewitt operator. In the literature, there are a range of suggested alignment methods used in similar experiments. These include focus feedback using Normalised Variance [30] and wavelet-entropy [31], and vision based contact detection [32]. In other research, the Prewitt operator is also found to be the best option for focusing biological cells [28]. Using the operator in the current research, it was found that the end effector and focus plane could be reliably aligned with accuracy of 2 μm. Similar systems using alternative functionsfor microscope based auto-focusing have achieved results of 2 μm and slightly less than 2 μm [30,33], suggesting this may be a practical limit for alignment using image focus. Alternative methods for locating objects under microscope have also been demonstrated, including using multiple cameras and calculating location based on disparity [15]. This and other methods may be considered as options if further improvement of alignment accuracy is required.

### 4.2. Contact Adhesion

The strategy developed in this research takes advantage of the dominant forces at micro-scale, in particular the variability observed between the dry and the liquid environments. Using adhesion forces as part of a strategy has also been demonstrated in previous research, where a 50 μm glass sphere was captured and released by rolling [34]. The influence of environmental conditions on adhesion forces has also been researched in a number of other papers, suggesting factors such as temperature and surface structure can make a significant difference [35,36]. An alternative strategy has been used, where folding grippers were developed to physically enclose cells [37]. The ability to capture 10 μm red blood cells is demonstrated. The difficulty using a strategy such as this is object release, which was not discussed in that study. The strategy demonstrated in our research does not require any active release mechanisms, and instead uses the liquid environment to release the oocyst. One of the significant features of the proposed system is to not fight against adhesion forces (most conventional contact micromanipulation systems are designed to fight against adhesion) but to utilise them during the capture and release tasks. During the capture task, adhesion between the end effector and oocyst is maximized by increasing the contact area between the two. Moreover, this task is realized in air where adhesion force is stronger compared to liquid environments. For the release task, that strong adhesion bond between the end effector and oocyst is overcome through utilizing surface tension and capillary action.

The proposed system can successfully pick up a micro-object in the air environment and it is suitable for our oocyst isolation task. However, if the target object must be in a liquid environment during the pick up task (for some kind of cells, this might be a requirement), the proposed system would not be an ideal method, nor would it be for work in which viable oocysts are required, because oocyst survival is significantly reduced after prolonged desiccation compared to other conditions [38]. DNA, however, will be stable for extraction and sequencing and relatively unaffected by the drying [39].

### 4.3. Comparison to Existing Systems

The earliest examples of successful robotic micromanipulation of biological cells down to 10 μm diameter were in the early 2000s [40,41]. Since then a number of cells of this size and larger have been manipulated. Ultrasonic fields were used to align HeLa cancer cells of 20 μm, which were then physically manipulated with some success [42]. To the best of our knowledge the smallest cells successfully picked and released using contact micromanipulation are cyanobacteria cells of 9 μm diameter [20]. This system used an electrothermal actuated gripper to pick, and rapid movement to release the cells, overcoming adhesion forces by maximising inertia and contact area with the base of the substrate. Rapid movement allowing accurate release has also been demonstrated for 13 μm 3T3 mouse cells [43], where high frequency vibration of end effectors was used.

The system developed in this current study has demonstrated reliable pick and release of 5 μm *Cryptosporidium* oocysts, with sufficient accuracy for single cell analysis. Although other examples in literature demonstrate more accurate release of cells, particularly using rapid movement for release, they do not demonstrate performance of their systems with cells of less than 9 μm diameter, where cell inertia and surface area is much smaller. These existing contact micromanipulation systems have also not demonstrated relocating the cell to a new environment.

In comparison to existing FACS systems for single cell isolation, the performance of this system is slower. While some FACS systems demonstrate throughput of 15,000 cells per day or higher [10], this contact micromanipulation system requires up to 455 s per cell. However, the use of contact micromanipulation offers benefits over an FACS type of system. FACS systems have been used on *Cryptosporidium* oocysts for SCA, but rely on high numbers of oocysts per gram of faeces [11]. When cell populations are very small—as is typically the case for *Cryptosporidium* oocysts procured from a water supply—the success rate of the method used ensures that the maximum number of available cells can be isolated and analysed. The success rate for this contact micromanipulation system of 98% is promising. An FACS system must also isolate cells in the same liquid environment, due to the non-contact method, whereas this system can isolate cells between two separate environments (dry and liquid).

## 5. Conclusions

The strategy chosen for this project has proven the ability to isolate *Cryptosporidium* oocysts of 5 μm diameter in a new environment. A physical system minimising vibration and control system designed to optimise accuracy was used to achieve the result of 98% success rate (49 out of 50 trials) for parasite separation task. Although other existing systems have proven impressive results with small objects and some have demonstrated the ability to accurately manipulate biological cells, to the best of our knowledge, this is the first contact micromanipulation system capable of isolating a cell of <10 μm diameter in a new environment. Future work for the system will be single cell analysis of the *Cryptosporidium* oocyst, potentially leading to an improvement in understanding of the parasite.

## Figures and Tables

**Figure 1 micromachines-11-00003-f001:**
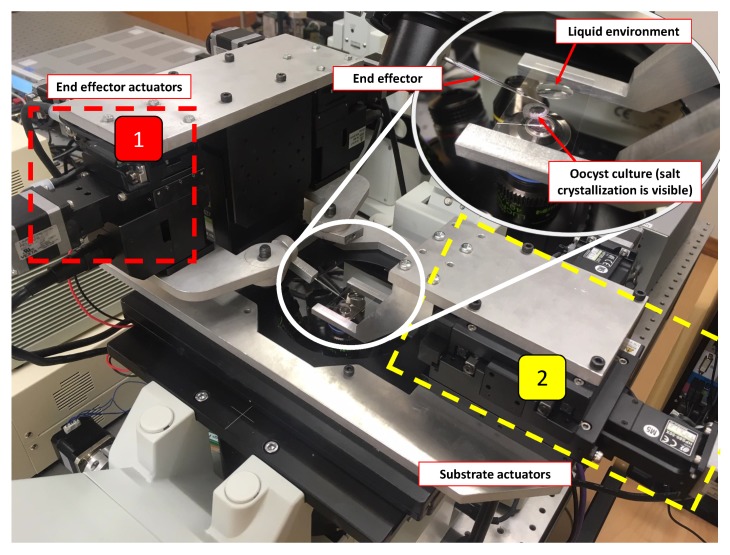
Physical system developed for task: (**1**) Local x,y,z actuators. (**2**) Substrate x and y actuators.

**Figure 2 micromachines-11-00003-f002:**
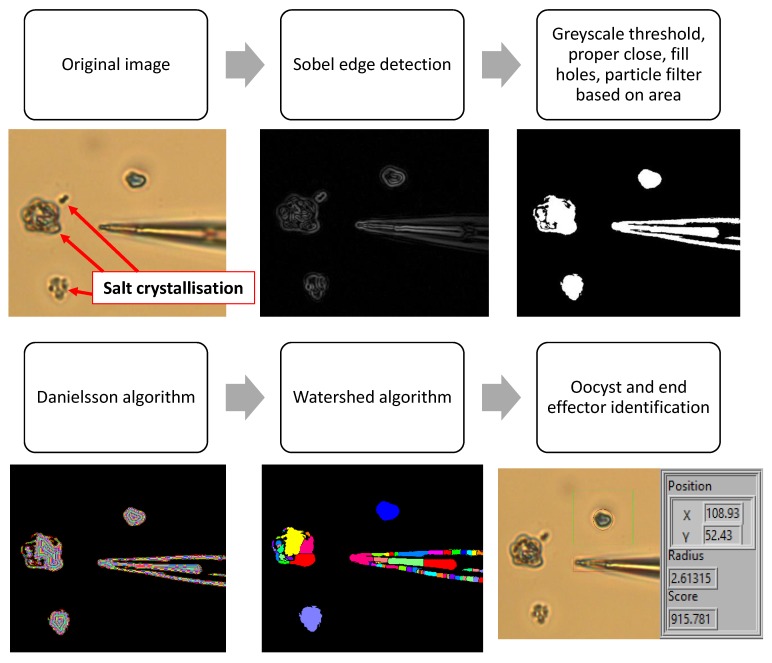
Image processing techniques used to find the tip of the end effector and the size and location of the oocyst.

**Figure 3 micromachines-11-00003-f003:**
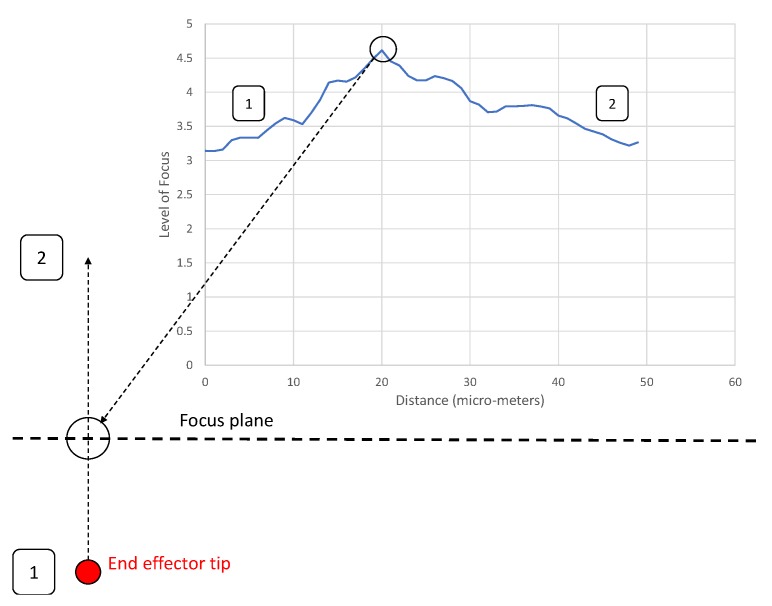
Process for alignment of the end effector and focus plane. End effector tip moves from point 1 to 2 in increments of 0.5 μm, then moves back to the point of peak focus.

**Figure 4 micromachines-11-00003-f004:**
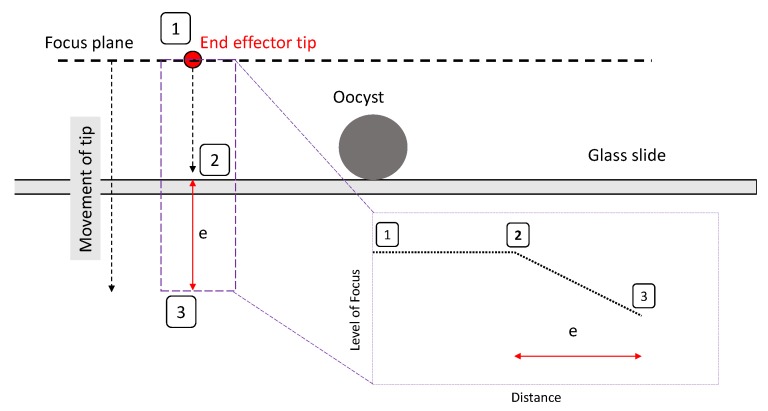
Process for alignment of the end effector with substrate.

**Figure 5 micromachines-11-00003-f005:**
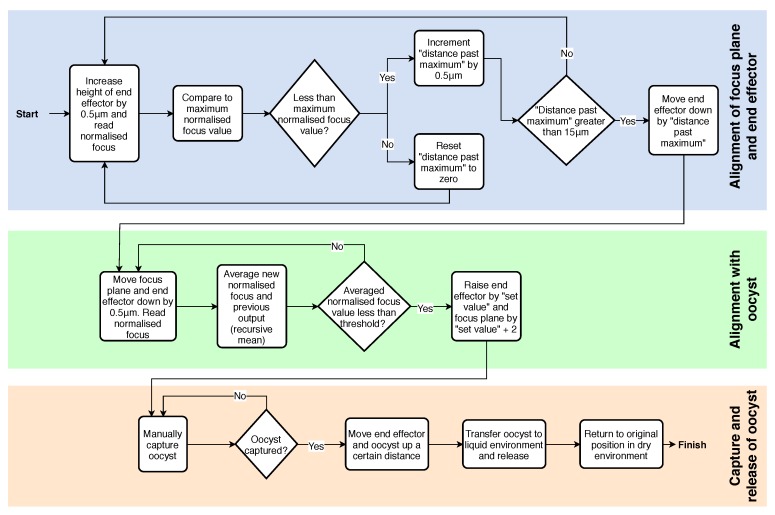
Flow chart representing full control algorithm for automated alignment and oocyst isolation.

**Figure 6 micromachines-11-00003-f006:**
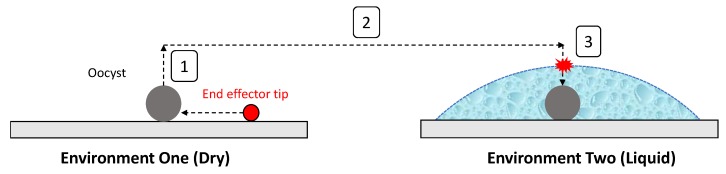
Strategy for automated relocation and release of the oocyst: (**1**) The end effector captures the oocyst using contact adhesion forces. (**2**) The oocyst is moved to the desired micro-well. (**3**) The end effector enters the liquid environment and the oocyst is released due to the capillary action.

**Figure 7 micromachines-11-00003-f007:**
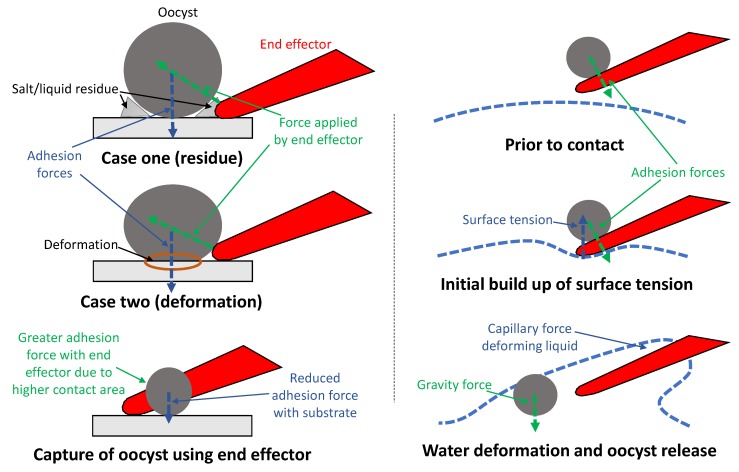
Physical mechanics of the capture and release strategy

**Figure 8 micromachines-11-00003-f008:**
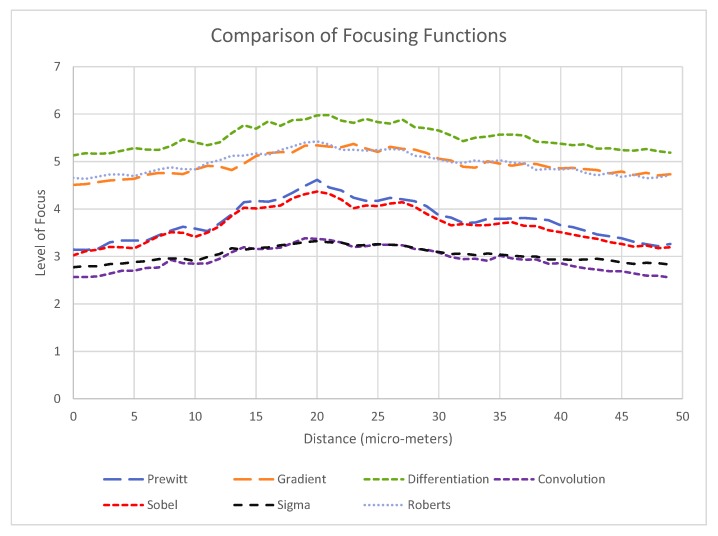
Level of focus readings for a range of edge detection operators for the same end effector being panned through the focus plane.

**Figure 9 micromachines-11-00003-f009:**
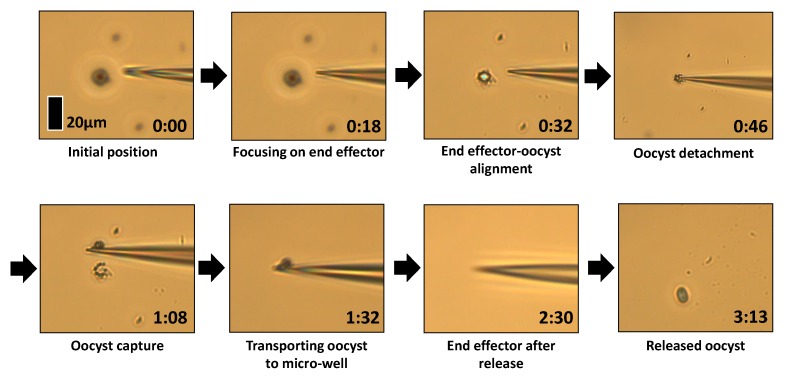
Full process of alignment, capture in the dry environment, and release in the liquid environment.

**Figure 10 micromachines-11-00003-f010:**
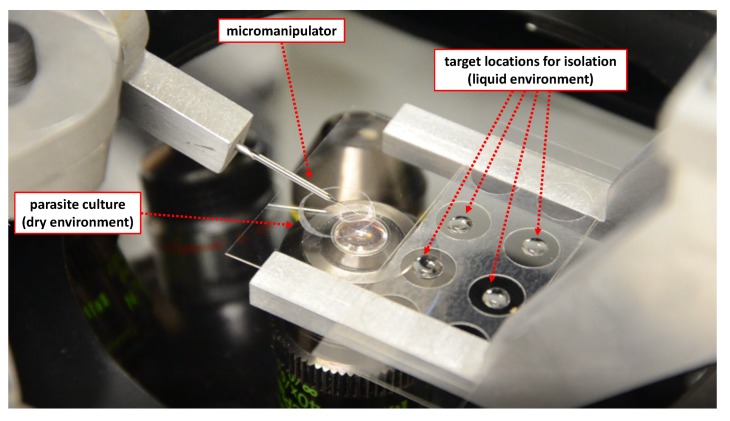
Demonstration of cell isolation capability of the proposed approach: four *Cryptosporidium* oocysts are picked from dry environment and placed in four different micro-wells (liquid environment).

**Table 1 micromachines-11-00003-t001:** Results of alignment testing using Prewitt operator.

Accuracy	Successes	Success Percentage (95% Confidence Interval)
<2 μm	27	90% (73%–98%)
<1 μm	15	50% (31%–69%)

**Table 2 micromachines-11-00003-t002:** Success rate for oocyst pick and place task

Successes	Success Percentage (95% Confidence Interval)
49/50	98% (89%–100%)

**Table 3 micromachines-11-00003-t003:** Duration of each sub-task for pick and place procedure.

Task	Average Time (s)	Maximum Time (s)
Alignment (end effector and focus plane)	21 (σ = 2.6)	25
Alignment (end effector and oocyst)	16 (σ = 2.4)	20
Manual capture	41 (σ = 50.2)	270
Oocyst release and return to origin	140 (σ = 0)	140
Total	218 (σ = 50.7)	455

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
