# Peer review of "Micromanipulation System for Isolating a Single Cryptosporidium Oocyst"

_micromachines, 2019, doi:10.3390/mi11010003_

Round 1
Reviewer 1 Report
In this manuscript, the authors present a partly automated system for contact micromanipulation of Cryptosporidium oocysts and claim the developed has a success rate of 98%. In general, the topic of this manuscript is quite interesting. The authors rely on the well-known imaging process and control algorisms for the demonstration. In the end, the authors claimed that there would be improvement of the developed manipulation system, in terms of a smaller size of the object. However, there are some of discrepancies on the experimental design and obtained results to support what was claimed. There are the following major concerns: First off, there is no control mechanism to release the object for precise localization. The proposed system is simply dependent on uncertain and unmeasured surface tension. To perform single cell analysis, functions for both pick-up and release should be controllable. The authors should elaborate the mechanism of pick-up and release of the cell. Secondly, the authors should have enough data sets to claim the results statistically valid. For instance, One-ANOVA, t-test, and normal distribution curve fitting should be performed to show a statistically difference. This is important data for such control system to represent reliability and reproducibility. The result and discussion are not supportive to the claim that authors made. Considering that Micromachines publishes considerably novel studies and findings in the microsystems and device, leading to novel applications, I would not recommend the publication of this manuscript on Micromachines. In addition, the following items need to be considered for corrections.
In lines 67-72, the paragraph seems redundant. Suggest removing it. No need to list up the contents in the last part of the Introduction section.
In line 84, there should be a representative/quantitative value for the method (e.g. what are the level length reduced and the stiffness increased?).
In lines 92-101, the paragraph should be revised in terms of tense.
In line 102-112, for consistency, suggesting writing a paragraph with all necessary information (company, city, nation, etc.), instead of bullet-style as part of manuscript.
In line 114, what DAQ device was used to develop the control system?
In figure 3, what is a unit for the level of focus?
In figure 6, again, there is no control of the release process, causing inaccuracy of localization of the cell. For instance, the end position can not to expected, while the cell falling down to the bottom of the glass surface. Moreover, there may be a chance to have the cell floating on the surface of the liquid drop. And what is the expected thickness of the water drop?
In line 169, the authors should elaborate what type of adhesion force is applied to pick up the dried cell.
In line 182, I do not see how it is ‘fully’ dried in 2-3 hours. The morphology of the cell could’ve completely deformed or ruptured, due to capillary force, if it is fully dried in air.
In line 191, I do not agree that 30 trials can give the conclusion. Need more trials to perform the statistical analysis.
In lines 193-202, there should be references for the criteria for effective operator that the authors define or follow.
In line 211-212, the statement that a 2 um accuracy is sufficient for the pick and place operation is subjective. How can 2 um be sufficient for accuracy?
In line 214, Statistical analysis is required.
In lines 225-226, without statistical analysis, it would be difficult to see if the single failure is significant or not.
In figure 8, scale bar is required. Need time information for the time lapse images.
In line 231, it sounds some parts of the cell were lost.
In line 233, image and detailed description are required for the moment to show how the cell attached to the end effect is transported across the air/water interface.
In table 3, there is no information from statistical analysis.
In line 237, how many samples were tested to get the required time (213 seconds)? Is it an average value?
In line 253, The authors should clarify that what similar systems are used and results in a similar range. The information should be specific.
In lines 266-268, the adhesion mechanism should be explained, which is critically relevant to the overall performance of the manipulation system for single cell analysis. Also, I wonder why the active (controlled) release mechanism is not required. There is no clear reason.
In line 286, how much slow is it?
In line 287, I wonder how the developed micromanipulation offers benefits over an FACS type of system. It can perform up to 363 cells/day vs. 15,000 cells/day with FACS system as mentioned.
In line 290, ‘becomes much more important’ sounds subjective and vague.
In lines 292-293, In general, cell viability is important parameter during single cell analysis. The authors may consider of cell viability and functions that are greatly affected by drying process in air.
In line 300, without presenting enough data and elaborating the adhesion/release mechanisms, I don’t follow the claim that isolating a cell smaller than 10 um would be new and novel for micromanipulation.
Reviewer 2 Report
The authors reported a semi-automated micromanipulation system, and they demonstrate the usage of the system in picking a single-celled microorganism from a dry substrate and placing them in a liquid environment. The system is capable of automatically aligning the microscope focusing plane with the tip of the micromanipulator and aligning the focus plane with the glass substrate on which the microorganism cell rests. A user has to manually pick the cell from the substrate and lift it above the substrate. Compared with previous work, the key novelty here is the small size of the cell being manipulated and also the pick-and-release strategy. The presentation of the work is clear and concise, and there are a few comments and suggestions that the authors need to address before acceptance for publication.
Why the cryptosporidium oocyst is chosen? It seems that this choice does provide some convenience in the demonstration of manipulation. For example, they are not motile. Does the pick-and-place manipulation damage the cells? Are these cells still alive after manipulation? What are the safety considerations when doing these types of experiments? It seems to be difficult to tell from optical microscope images alone whether a particular particle is a cell or some other random dust particles of similar size. How can a user make sure these are cells? Would some kind of fluorescence imaging help? At the stage where the cells are picked up from the substrate, does the tip of the manipulator only move in XY direction, or in z-directions as well? May the tip be broken? Is the liquid environment into which the cell is released aqueous solution? What is the nature of the adhesion between the micromanipulator and the cell? How does it differ from the interaction between the cell and the dry glass substrate? The authors need to address the limitations of the reported pick-and-release strategy. The dry environment does significantly help the task of picking, but not all cell-based manipulation can be done in a dry environment. The times listed in Table 3 needs to include standard deviations.Author Response
Please see the attachment

Reviewer 3 Report
Dear Authors,
the paper represents a topic of interest for the community. However, the structure of the manuscript must be improved and some parts must be presented more clearly. The method and the related steps for its implementation must be better discussed.
Here are my comments:
Section 1
- Concerning reference [5], why the cell isolation within the same liquid environment represents an issue?
- Which are the limits or disadvantages of the work in [12]?
- You stated that [19] does not demonstrate the isolation of a cell in a new environment. What is the need for isolating it in a new environment?
- What is the novelty of your method? Other works report of focusing the gripper and target object for the scope of manipulation or assembly, then what is the originality here? Is it related to the application of the method to the manipulation of this parasite? Or to the fact that you manipulate in different environments? Or both? However, it not clear to me why there is the need for taking the sample from water, letting it dry, then picking it to release it in liquid again. Please provide a clear context and objectives.
Section 2
- Why are five degrees of freedom necessary? Why not e.g. 4 or 6?
- Figure 1: Please add labels to easily identify the different devices of the experimental setup. In the detailed view, is there a drop of water (or other liquid environment to be specified) on the substrate? Does it emulate a well of a multi-well plate? What is the spot or stain where the oocytes are picked up? Is it debris? To what is this due?
- How are the Sigma Koki stepper motors distributed? Which model for which axes since they are not all the same?
- Please specify which is the point where vibration does not have any significance impact on the performance of the system (line 98).
- In general, it is useful to provide all the models of the devices you used, however most of them now appear as a list of products. This part should be improved describing how they are connected each other for the functionality of the modules and of the whole station. The product models in 2.2.1 should be included in the description (for example in 2.2 and partly in 2.3). Which automated focusing stepper motor system was used (model missing, line 107)? Which model of Thorlabs table was used and what was the air compressor used for? Which are the technical data of the IDS uEye camera and of the complete vision system (Sensor size and type, pixel resolution, field of view, spatial resolution, working distance, etc.)? Did you use commercial testing spheres or did you manufacture them?
- In 2.3, you stated that the feedback from a microscope camera was used. Which strategy did you use? Look and move, visual servoing (PBVS or IBVS), or others? Please provide more details about the method.
- Figure 2: with reference to the bottom right sub-image, which is the unit of measurement of position and radius (of the oocyte?)? Pixels or mm? How was the vision system calibration performed? Is the “score” referring to the match score of the detected oocyte with an oocyte template? Which matching method did you use (pattern matching or geometric matching? Edge-based or Feature-based?)
- Lines 135-136: Is the oocyte to be picked in a fully dry environment or is it covered by a film of water or other liquids?
- Figure 3: how is the level of focus defined? Which is the unit of measurement? What “0” represents in the x axis? Why increments of 0,5 microns were chosen?
- Line 143: How was the threshold value experimentally derived? And how was the distance (1 mm) for lifting the end effector back up derived?
- Figure 6: on the left, shouldn’t the end effector touching the substrate (as said in line 159)? On the right, how is the entrance in the drop? Does the end effector enter at an angle (e.g. 45°)? How is the contact? Is it necessary to push? Does the oocyte deform or stress? Does the oocyte sink in the liquid or float?
- Line 168: did you consider automating the oocyte capture? Why manually? Which are the challenges or issues related to the automation of this step?
- Line 171-172: did you directly observe the magnitude of the force relying on the surface area in contact or is it something derived from literature? What do you mean with “physical disturbance” of the oocyte? Did you act only empirically or do you study or analyse a model for this manipulation strategy and the interaction forced to allow for the oocyst picking? Did you rely on a theoretical approach and simulation? I think this could be significant for the quality of the work, and the effectiveness and reliability of the method.
- Lines 176-177: a better explanation of your observation must be provided.
- At the beginning, when picking the first oocyte, the gripper tip is dry. After the release I expect the tip to be wet. This influences the adhesion forces between tip and sample, there the manipulation strategy. How did you take into account this aspect?
Section 3
- Lines 181-183: why do you take the sample from water, let it dry, then put it in water again? Why not manipulating in liquid-to-liquid environment?
- Line 190-193: why 30 trials? Is there a statistical significance? How were the experiment run? Please provide a scheme or a picture.
- Line 202, last sentence: what about repeatability then?
- Line 210-212: this sentence should be moved before, when citing the criteria 2 for an effective operator.
- Table 1: how is accuracy defined? Which are angles and dimensions of the end effectors? Concerning the success, what happened in the 3/30 and 15/15 cases of no success?
- Line 214: why did you performed 50 trials? Has this number a statistical significance?
- Lines 233-234: there are concept repetitions. Please avoid repetitions if not necessary.
- Table 3: the manual capture time can vary a lot (maximum time almost double then average time), e.g. depending on the operator skills. Who executed the tests? How is the trend of the time needed to perform this operation? Did you observe a reduction of time because of acquired experience as the trial number increased or an increase of time due the tiredness?
- Lines 237-239: How should the oocytes be stored and maintained? Do they suffer out of liquid? Do they need special environmental conditions? How long can they resists? Could they die or being damaged?
Section 4
- The discussion reports many works in literature that should appear at the beginning of the manuscript, together with the rest of the state of the art. Please move them in section 1 and focus your discussion to your method beyond the state of the art.
- Lines 252-253-274: how was accuracy defined?
- Lines 259-260: a theoretical approach and model are missing.
- Lines: 280-283: How much does it change from 9 microns to 5 microns?
- Line 290-291: how is reliability defined? could the sample suffer from this manipulation method or special attention should be paid?
- References: some references are older than 20 years. Please consider substituting them with more recent works if pertinent.
Best regards
Round 2
Reviewer 1 Report
The manuscript has been significantly improved. The authors have provided some of the important points in the discussion section. Therefore, I would the publication of the revised manuscript on Micromachines. There are a couple of concerns the authors may consider correcting.
Lines 191-201, suggest checking out the tense of the statements. Both past and present tenses are used together.
Line 373, suggest adding references to support the statements.
Author Response
Respond to Reviewer 1's Comments
General Comment: The manuscript has been significantly improved. The authors have provided some of the important points in the discussion section. Therefore, I would the publication of the revised manuscript on Micromachines. There are a couple of concerns the authors may consider correcting.
Answer: We thank the reviewer for the encouraging comments. We will answer questions in green colour and make the changes to the manuscript in green colour so the reviewer can differentiate current changes (in green colour) from previous changes (red colour).
Q.1. Lines 191-201, suggest checking out the tense of the statements. Both past and present tenses are used together.
Answer: This has been corrected now (see lines 195 - 205).
Q.2. Line 373, suggest adding references to support the statements.
Answer: We have added the necessary reference (DNA extractions from dried oocysts, but from fecal samples not water) in lines 386 - 387.

Reviewer 3 Report
Dear Authors,
I think other revisions are necessary to the manuscript. Here are my comments:
-In the water drop simulating a well of a multi-well plate? Or are the oocytes kept in the water drop during cracking and for further analysis? If the drop is only an example of release environment, it should be considered that the actual shape of the liquid surface will be different then the release will be different and the strategy change. Please provide a clearer explanation and possible comments about this aspect.
-The explanation about the presence of salt particles in the image must be provided in Fig. 2 that is the first time the reader sees them.
-In particular in the introduction, when saying “new environment” please specify which environment, for example citing it in brackets.
-You could avoid citing twice ref 13 in lines 39-41.
-More details should be provided about the vision system calibration. Moreover, what about noise filtering and image distortion correction? Information on the image resolution, field of view, etc. are still missing.
-It is still not clear to me how accuracy was defined. The mathematical relation computed for accuracy calculation should be provided (lines 175-187).
-The time calculation was based only on 5 random trials, but it must be performed based on all trials. Comments to the obtained results should be added, similar to those the authors did to reply to this reviewer in Q.28.
Best regards
Round 3
Reviewer 3 Report
Dear Authors,
I think a few revisions are still needed before publication:
1) First of all, in the document you provided as supplementary material to explain the accuracy, it is important to clearly state which is finally the quantity that you declare to be the “accuracy” (the error “e”?). Moreover, you should add to the figure the reference frame to clearly identify x, y, z directions. Indeed, in the text you explain the model by referring to these directions, but then they are not visible in the figure.
2) The spatial resolution is more commonly expressed in micrometers/pixel rather than in pixels/micrometers. Please modify it at line 150.
Thanks,
best regards